# Association between an Increased Serum CCL5 Level and Pathophysiology of Degenerative Joint Disease in the Temporomandibular Joint in Females

**DOI:** 10.3390/ijms24032775

**Published:** 2023-02-01

**Authors:** Haruhisa Watanabe, Takashi Iori, Ji-Won Lee, Takashi S. Kajii, Aya Takakura, Ryoko Takao-Kawabata, Yoshimasa Kitagawa, Yutaka Maruoka, Tadahiro Iimura

**Affiliations:** 1Department of Oral Diagnosis and Medicine, Faculty and Graduate School of Dental Medicine, Hokkaido University, Sapporo 060-8586, Japan; 2Department of Pharmacology, Faculty and Graduate School of Dental Medicine, Hokkaido University, Sapporo 060-8686, Japan; 3Department of Orthodontics, Keiyu-kai Sapporo Hospital, Sapporo 060-0061, Japan; 4Pharmaceuticals Research Center, Asahi Kasei Pharma Corporation, Izunokini 410-2321, Japan; 5Department of Oral Surgery, Center Hospital, National Center for Global Health and Medicine, Tokyo 162-8655, Japan

**Keywords:** CCL5, temporomandibular joint, degenerative joint disease, mandible, joint, bone metabolism, osteoarthritis

## Abstract

Degenerative joint disease of the temporomandibular joints (DJD-TMJ) clinically manifests with symptoms such as orofacial pain, joint sounds and limited jaw movements. Our research group previously reported the functional necessity of a chemokine-chemokine receptor axis of CCL5-CCR5 in osteoclasts. Accumulated studies reported that this axis was involved in the pathogenesis of bone and joint destructive diseases, suggesting CCL5 as a potent biomarker. This study investigated whether or not the serum level of CCL5 can be a biomarker of DJD-TMJ and concomitantly analyzed changes in the serum and urine levels of bone markers to see whether or not changes in the rate of bone metabolism were predisposing. We enrolled 17 female subjects with diagnosed DJD-TMJ and sexually and age-matched 17 controls. The serum CCL5 level in DJD-TMJ subjects was significantly higher than that in the control subjects. Multivariate analyses indicated an association between an augmented CCL5 level and the rate of bone metabolism, especially in relatively young DJD-TMJ subjects without other systemic symptoms. A principal component analysis of serum markers and our pharmacological experiment using a postmenopausal model of ovariectomized rats suggested that an augmented serum CCL5 level specifically reflected DJD-TMJ and that covert changes in the rate of bone metabolism predisposed individuals to DJD-TMJ.

## 1. Introduction

Degenerative joint disease of the temporomandibular joints (DJD-TMJ) is characterized by pathomorphological changes in the condyle and articular eminence [1,2,3], with clinical manifestations of symptoms such as orofacial pain, joint sounds and limited jaw movements. If DJD-TMJ occurs during adolescence, there may be interference with normal condylar development, leading to mandibular asymmetry, retrognathia and anterior open bite [4].

The TMJ disc uniquely functions as a lubricator and a load distributor over the condylar surface [5]. TMJ disc displacement has been proposed to be associated with the onset and progression of DJD-TMJ, which interferes with condylar mobility, leading to the overloading of the anterior surface of the condyle [6,7]. Subsequent destruction of articular cartilage and subarticular bone can occur over time in DJD-TMJ [8,9]. Therefore, irregular loading, such as joint trauma, parafunction, occlusal instability, functional overloading and increased joint friction, can be involved in the local etiology of DJD-TMJ [4,10].

The predisposing etiology of DJD-TMD is multifactorial [4,10], and its predisposing factors include systemic diseases, such as rheumatoid arthritis, systemic lupus erythematosus (SLE), low estrogen levels and steroid usage, as well as iatrogenic factors, such as orthodontic treatment and orthognathic surgery [11]. Possibly due to this multifactorial etiology, conceptual frameworks of diagnostic definitions are still controversial and overlap with each other to some extent.

Currently, radiographic examination is an effective method of visualizing the degree of cartilage and bone loss for a diagnosis of DJD, but it is not a predictive tool. Therefore, biomarker research is expected to be useful for developing tests that are predictive rather than reactive [12]. Biomarkers can be found in biological fluids, including serum, urine and synovial fluid. Serum is the liquid portion of blood. Since immune reactions occur primarily in the blood, serum biomarker research has been primarily focused on cytokines, chemokines, proteins and miRNA molecules. Urine contains waste products of plasma filtration, which reflects the current physiologic state of the body. Urine can provide large volumes of biological fluids with noninvasive techniques. Synovial fluid is a biological fluid used for investigating the progression of DJD, as changes in the joint tissue environment will directly affect synovial fluid composition. However, sample collection from the TMJ requires arthrocentesis due to the small volume of such biological fluid, a technique that can be invasive and may affect the reproducibility of the obtained data [12,13].

Biomarker research requires a pathophysiological understanding at the molecular level. The pathophysiology of DJD-TMJ, however, remains largely unclear. Changes in levels of cytokines, chemokines and other growth factors are reportedly involved in osteoarthritis (OA) [14]. Factors involved in inflammation and immunity can be detected from the synovial fluid of the TMJ [15,16]. Chemokines, including C-C motif chemokine ligands (CCLs), were reported to be involved in degenerative changes in the knee joint cartilage [17]. Chemokines are a family of small proteins. Through their G protein-coupled receptors, such as C-C chemokine receptors (CCR), they play roles in the migration and infiltration of immune cells. Our previous researches revealed that chemokine/chemokine receptor systems play essential roles in bone metabolism and osteoporosis [18,19,20]. Therefore, chemokines are likely to be mediators of inflammatory and bone/cartilage-destructive diseases. We previously observed that mature osteoclasts highly expressed C-C chemokine receptor type 5 (CCR5) and that CCR5 and C-C motif chemokine ligand 5 (CCL5), a major ligand for CCR5, were involved in bone-destructive diseases, such as osteoporosis and rheumatoid arthritis [20]. Feng et al. reported that the CCL5 level was significantly higher in the synovial fluid of the TMJ in DJD-TMJ patients than that in healthy controls, suggesting the involvement of CCL5 in the pathophysiology of DJD-TMJ [21].

These findings prompted us, in this study, to assess whether or not the serum level of CCL5 can be a biomarker of DJD-TMJ. We enrolled 17 female subjects with diagnosed DJD-TMJ regardless of other diseases and sexually- and age-matched 17 healthy controls. We also concomitantly investigated changes in the serum and urine levels of bone markers to see whether or not changes in the rate of bone metabolism predisposed individuals to DJD-TMJ.

## 2. Results

### 2.1. Elevated Serum Levels of CCL5 in Whole DJD-TMJ Patients

We first compared the serum levels of CCL5 in the DJD-TMJ group of 17 total subjects and the healthy control group of 17 total subjects. The DJD-TMJ group showed a significantly increased level of serum CCL5 compared to the control group (Figure 1a and Table 1).

### 2.2. Correlations of Serum CCL5 with Bone Markers and Inflammatory Markers Whole DJD-TMJ Subjects

We then conducted a multivariate analysis to see how these markers correlated with each other (Figure 2). We found that, in the DJD-TMJ group, the CCL5 level showed a moderate positive correlation with the bone resorption markers of urine NTX, CTX and DPD, an anabolic bone marker serum BAP, and serum TNF-α, while in the control group, the CCL5 level showed weak negative correlations with these bone markers and TNF-α. Levels of CCL5 showed a greater negative correlation in the DJD-TMJ group than in the control group. These data suggested that the augmented level of CCL5 in the DJD-TMJ DJD-TMJ group was associated with increased bone turnover and an inflammatory marker.

### 2.3. Correlations of Serum CCL5 with Bone Markers in the Younger DJD-TMJ Subjects

We previously reported that the incident rate of DJD-TMJ was bimodal, being highest among females in their teens to 30s, followed by those in their 50s and older [3], which prompted us to divide the patients and the controls into two groups according to age. The average age of all DJD-TMJ subjects and the control subjects was 42 years old. Therefore, we established a younger group of subjects at ≤42 years old and an older group at >42 years old and compared each group with the corresponding age-matched control group (Figure 3 and Figure 4, Table 2 and Table 3, Appendix A). The serum CCL5 levels in the older DJD-TMJ group were significantly higher than in the older control group, while in the younger DJD-TMJ group, the serum CCL5 level showed a tendency to be higher, albeit not significantly, than in the younger control group (Figure 3a and Figure 4a). The levels of TRAP5b, CTX/Cr and DPD/Cr in the younger DJD-TMJ group were significantly higher than in the younger control group (Figure 3b). In the older DJD-TMJ group, the urine CTX level, but not CTX/Cr, was significantly lower, while the TNF-α level was higher, albeit not significantly, than those in the age-matched control group.

Our multivariate analysis interestingly showed different tendencies between the younger and older groups (Figure 5). In the younger DJD-TMJ group, the levels of CCL5 and bone markers, especially CTX and DPD/Cr, showed strong positive correlations (Figure 5a). However, the older DJD-TMJ group did not show an obvious correlation, while the TNF-α level had a positive correlation (Figure 5b). These analyses suggested that the pathophysiology of DJD-TMJ may differ by age; bone metabolism was suggested to be related to the pathophysiology in the younger DJD-TMJ subjects but not in the older DJD-TMJ subjects.

### 2.4. Independency of the Serum CCL5 Level from Changes in Bone Turnover Markers and Female Hormones

We further conducted a principal component analysis on the measured serum and urine markers of all control and DJD-TMJ subjects (Figure 6a). The result of this analysis showed that serum levels of CCL5 and BAP may have affected each other. Our Pearson’s analysis in younger DJD-TMJ subjects, but not in healthy subjects, showed that CCL5 and bone metabolism markers have a positive correlation. These findings suggested the necessity of further biological study between CCL5 and bone metabolism.

To experimentally confirm whether or not the serum level of CCL5 was affected by changes in bone turnover, we utilized ovariectomized (OVX) female rats as a model of postmenopausal osteoporosis; they were treated with an anti-osteoporotic drug of teriparatide (TPTD), an active fragment of human PTH, of which intermittent administration pharmacologically stimulates bone turnover in a bone anabolic manner (Figure 6b). The result showed that serum CCL5 levels did not show a significant difference between sham-operated control and OVX rats. Treatment of OVX rats with TPTD did not cause significant changes in the serum levels of CCL5. This model animal experiment suggested that elevated levels of serum CCL5 in the DJD-TMJ subjects were likely to be associated with the pathological condition of the TMJ, not only due to changes in bone turnover or a decreased female hormone level.

### 2.5. Serum CCL5, a Potent Biomarker of DJD-TMJ

To evaluate whether or not the serum CCL5 level can be used as a biomarker of DJD-TMJ, we drew the receiver operating characteristic (ROC) curves of all patient and control groups, including the selected patient and control groups, the young patient and control groups and the old patient and control groups (Figure 7). The results indicated that serum CCL5 can be used as a biomarker for all DJD-TMJ subject groups with moderate accuracy (area under the curve [AUC] = 0.7093), for the younger group with mild accuracy (AUC = 0.6636) and especially for the older group with moderate accuracy (AUC = 0.7857), suggesting that serum CCL5 is a potentially useful marker of DJD-TMJ. However, the pathophysiology of DJD-TMJ seems to differ by age.

## 3. Discussion

We observed that the serum CCL5 level in the DJD-TMJ subjects was significantly higher than that in the healthy control subjects. Characteristic features of DJD-TMJ are considered to be initial damage and degeneration of cartilage and subsequent pathological changes in subchondral bone and synovium. It was, however, reported that an increased turnover rate of subchondral bone can occur prior to cartilage degeneration in other portions of DJD, such as in the knee and hip [22,23]. Based on clinical observations, the active stage of osteoarthritic condylar deformity in DJD-TMJ was expected to take one to four years to reach a stable end state [24,25,26,27]. Using cone beam micro-computed tomography, Feng et al., radiographically divided subjects with recent onset (within one year) of disc displacement without reduction (DDw/oR) into two subgroups (those without and with DJD-TMJ) and reported that the average duration of DDw/oR without and with DJD-TMJ was 1.3 and 4.8 months, respectively [21]. Furthermore, their cytokine array screening in synovial fluid obtained from TMJ identified a significant increase in the synovial CCL5 level in DDw/oR subjects with early-stage DJD compared to healthy control subjects. Together, these present and previous findings suggested that CCL5 levels in serum, as well as synovial fluid, reflect the pathophysiology of DJD-TMJ that could also be associated with an increased bone turnover rate.

CCL5 is reportedly produced by joint-composing cells, such as chondrocytes, synovial fibroblasts and immune cells [28,29]. In rheumatoid arthritis and OA, upregulation of CCL5 levels in serum or synovial fluid has been reported, which may participate in the migration of monocytes, T cells, natural killer cells and eosinophils [30,31]. CCL5 also plays roles in the infiltration of immune cells and the stimulation of cytokine production in synovial fibroblasts [14]. Elevated production of CCL5 in chondrocytes and synovial fibroblasts induced osteoarthritic cartilage degeneration through the induction of MMP1, MMP3 and MMP13 [32,33]. We previously reported that CCL5 was an essential regulator of osteoclast differentiation and function through its receptors CCR1 and CCR5, respectively [18,20]. Therefore, upregulated CCL5 is likely to be involved in the degeneration of articular cartilage and subchondral bone. Furthermore, we previously observed that circulating CCL5 in blood was essential for normal osteoclast function and bone metabolism through observations of mice inoculated with neutralization antibodies against CCL5 [20]. The elevated serum CCL5 level in our subjects with DJD-TMJ may have suggested the pathological stimulation of osteoclast-genesis and its function.

Our multivariate analysis indicated that relatively active bone metabolism was associated with an increased CCL5 level in the younger DJD-TMJ group. It has been reported that an extremely low serum 17b-estradiol level due to oral contraceptive pill use and abnormal menstrual cycles in women is a major factor in PCR [34]. It is well-established that sex steroids play critical roles in the maintenance of healthy bone metabolism [35]. A sudden decrease in female hormone levels due to menopause reportedly induced the production of inflammatory cytokines, including IL-1b, IL-6, TNFα and M-CSF in bone marrow stromal cells, lymphocytes and macrophages; moreover, it augmented the expression of the receptor activator of nuclear factor-kappa B ligand (RANKL) to stimulate osteoclast-ogenesis [36,37]. A decreased bone volume in subchondral bone of the TMJ has been reported in postmenopausal OVX animal models [38,39]. In contrast, a high level of estrogen also reportedly deteriorated DJD-TM, which may also have affected the bone metabolism rate [40,41]. Therefore, it is possible that changes in estrogen levels, even in the younger subjects, can affect the levels of bone markers, as observed in this study. However, any of the younger DJD-TMJ patients in this study showed detectable levels of serum b-estradiol that were comparable to those in age-matched controls, and none of them were taking oral contraceptive pills, though serum sampling timing was not fully considered in terms of the menopausal cycle. OVX- (thus, estrogen-deficient)-rats in this study did not affect the level of CCL5. Our findings here suggested that the covert increase in the rate of bone turnover may have predisposed individuals to the occurrence of DJD-TMJ, despite no symptomatic signs of osteoporosis, including estrogen deficiency or other bone/cartilage degenerative diseases being observed, and that the higher tendency of CCL5 level in the DJD-TMJ was likely to be associated with a degenerative condition of temporomandibular joints.

Our multivariate analysis in older DJD-TMJ subjects showed that their increased serum CCL5 level was associated with the inflammatory marker TNF-α. While this group included subjects with autoimmune diseases, such as Sjögren’s syndrome and rheumatoid arthritis, and skeletal diseases, such as osteoporosis and multiple OA, their increased TNF-α level might have reflected a degree of inflammation that could have been a predisposing factor to DJD-TMJ, as with other age-related diseases [42].

Clinically, overloading and increased joint friction are well-accepted as being involved in the pathophysiology of DJD-TMJ [4,10]. Mouse models of DJD-TMJ show that overloading due to malocclusion induces cartilage destruction and changes in the turnover rate of subchondral bone, which involves apoptosis of chondrocytes and the induction of inflammatory cytokines [43,44,45]. Wu et al., using OVX mice with imbalanced occlusion, demonstrated that the combination of estrogen deficiency and excessive mechanical stress aggravated OA changes in the TMJ [46]. Their study also suggested that phosphorylation of ERK might be involved in crosstalk signaling between estrogen deficiency and mechanical loading. Along this line, we previously observed that CCL5 induced phosphorylation of ERK as well as Src in differentiating osteoclasts [18,20], suggesting the possible involvement of CCL5 in this crosstalk.

Through blockade experiments, CCR5, a critical receptor of CCL5, was shown to be essential in the functional differentiation of human osteoclasts [20]. The CCL5/CCR5 axis was previously shown to be involved in cartilage degeneration of clinical specimens obtained from knee joint OA [47,48]. A high expression of CCL5 was detected in cartilage specimens obtained from knee joint OA, while the value was below the detectable level in corresponding healthy specimens [48]. Furthermore, inflammatory signals, such as IL-1b and IL-18, induced CCL5 expression in cultured human cartilage. CCR5-deficient mice were tolerant to the development of knee joint OA [48]. These findings suggested that blockade of the CCL5/CCR5 axis may be a pharmacological target against the progression of DJD-TMJ through the inhibition of multiple aspects of cartilage/bone destruction.

The diagnostic criterion of idiopathic condylar resorption (ICR) and progressive condylar resorption (PCR) has been reportedly occurring particularly frequently in adolescents and is observed more in females than in males [49,50,51,52]. The prevalence of temporomandibular joint osteoarthritis (TMJ-OA) was reportedly higher in the elderly than in younger individuals, which can be conceptually defined as a skeletal disease associated with aging, as with knee OA [53,54]. In our previous survey, DJD-TMD was observed as highest among females in their teens to 30s, followed by those in their 50s and older [3]. This study enrolled female subjects with diagnosed DJD-TMJ regardless of other diseases; therefore, we assessed patients showing advanced degenerative changes of the TMJ with multiple etiologies. The age distribution of DJD-TMD subjects in this study also follows the previous reports (Appendix A). It is likely that the younger and older subjects include ICR/PCR and TMJ-OA, respectively, whose possible etiologies are discussed above.

This study has some limitations due to being a retrospective study. First, the relationship between the severityand detailed symptoms of DJD-TMD and the level of CCL5 was not able to be assessed because it is difficult to clinically manifest the initial symptom of DJD-TMJ due to its spontaneous and acute progression. In fact, most of the DJD-TMJ in this study are likely to be “advanced” DJD-TMD. Second, the prognosis of the patients and the change in the serum level of CCL5 were not assessed. A longitudinal study will reveal the relation between the pathophysiological condition of DJD-TMJ and the serum level of CCL5. Third, although our statistical analysis revealed several correlations between the levels of CCL5 and several markers, which could provide clues to figure out the causative pathophysiology of DJD-TMJ, further biological, as well as clinical research, will be required. Finally, the sample size in this study was small because of the difficulty in collecting a large cohort of DJD-TMJ patients [52], so it cannot exclude the possibility that the correlations we observed may be unstable. Although the incident rate of DJD-TMJ is quite low, further analysis using a large sample with a multicenter study will be desired.

Conclusively, the findings from the present study suggest that the augmented level of serum CCL5 was associated with the pathophysiology of DJD-TMJ. An increased rate of bone metabolism is also suggested to be a useful predictive risk marker of DJD-TMJ especially in relatively young subjects.

## 4. Materials & Methods

### 4.1. Clinical Subjects

All participants provided their written informed consent. The study was approved by the institutional review board of Center Hospital of the National Center for Global Health and Medicine (Classification: Non-specific clinical research, Approval No.: NCGM-G-003490-00) and the Clinical and Epidemiological Research Ethics Review Committee in Faculty and Graduate School of Dental Medicine, Hokkaido University (Approval No. 2020 No. 2).

DJD-TMJ was diagnosed using history, physical examination and radiographic examination [2]. The diagnostic criteria of DJD-TMJ were as follows: (1) spontaneous and acute regressive changes in dental occlusion and (2) the presence of radiographically marked deformity and resorption of the mandibular condyle, even without subjective symptoms (Figure 8 and Appendix A). We enrolled 17 female subjects with DJD-TMJ diagnosed from 2009 to 2020 at the Center Hospital of the National Center for Global Health and Medicine. Among the 17 DJD-TMJ subjects we analyzed in this study, six subjects had systemic diseases such as osteoporosis, autoimmune and related diseases, highlighted in Table 1, Table 2 and Table 3. Blood and urine samples were collected from the subjects at their first or later visit before giving any medical or surgical treatments. As healthy control subjects, we also enrolled 17 healthy women without TMJ symptoms or abnormal findings of mandibular head resorption, whose ages matched those of the DJD-TMJ subjects.

Serum was centrifuged at 2500 rpm for 10 min after blood collection, and the supernatant was collected and stored at –80 °C. CCL5 levels in serum were measured by an enzyme-linked linked immunosorbent assay (ELISA; Human CCL5 ELISA Kit, KE00093; Proteintech, IL, USA).

Levels of serum and urine bone metabolism markers and inflammatory markers were measured by a contracted company (SRL, Tokyo, Japan). The following is a list of the contracted analysis items:Bone resorption markers○CTX/Creatinine equivalent (CTX/Cr)○Urinary type I collagen cross-linked C-terminal telopeptide (Urine NTX)○Serum tartrate-resistant acid phosphatase (TRAP5b)
Osteogenic markers○Serum bone alkaline phosphatase (BAP)○Serum osteocalcin (OCN)
Inflammatory markers○Serum tumor necrosis factor-alpha (TNF-α)

### 4.2. Animals and Experimental Design

All experimental protocols were approved by the Experimental Animal Ethics Committee at Asahi Kasei Pharma Corp (Tokyo, Japan) and conducted in accordance with established guidelines concerning the management and handling of experimental animals. Three-month-old female Sprague-Dawley rats (Charles River, Yokohama, Japan) were housed in a dedicated laboratory animal facility with a 12-h light/dark cycle and unrestricted access to tap water and food (CRF-1; standard diet of rats; Oriental Yeast, Tokyo, Japan).

At six months old, the rats were randomly assigned to one of the following body weight-matched groups: sham ovariectomy (Sham) group (n = 8), ovariectomy (OVX) group (n = 8) or OVX-teriparatide administration (OVX-TPTD) group (n = 8); after assignment, they underwent bilateral ovariectomy or sham ovariectomy as appropriate. Two months after the operation, saline or 6.0 μg/kg teriparatide was subcutaneously injected 3 times/week for 4 months. Serum samples were obtained by centrifugation of blood samples collected from the subclavian vein in the morning on the last administration day. The CCL5 levels in the rat serum were measured using a Mouse/Rat CCL5/RANTES Quantikine ELISA Kit (R&D Systems, Minneapolis, MN, USA).

### 4.3. Statistical Analyses

Statistical analyses were performed using GraphPad Prism ver. 9.4.1 (GraphPad Software, CA, USA). All data were assumed to follow a normal distribution. The *t*-test was performed with the F test, followed by Student’s *t*-test for equal variances and Welch’s *t*-test for non-equal variances. The significance level was set at α = 0.05, and two-tailed tests were performed. The Pearson test and principal component analysis (PCA) were performed as a multivariate analysis. The results of the correlation coefficient analyses were interpreted and described as follows according to Schober P. et al. [55],

0.00–0.10: Negligible correlation

0.10–0.39: Weak correlation

0.40–0.69: Moderate correlation

0.70–0.89: Strong correlation

0.90–1.00: Very strong correlation

## Figures and Tables

**Figure 1 ijms-24-02775-f001:**
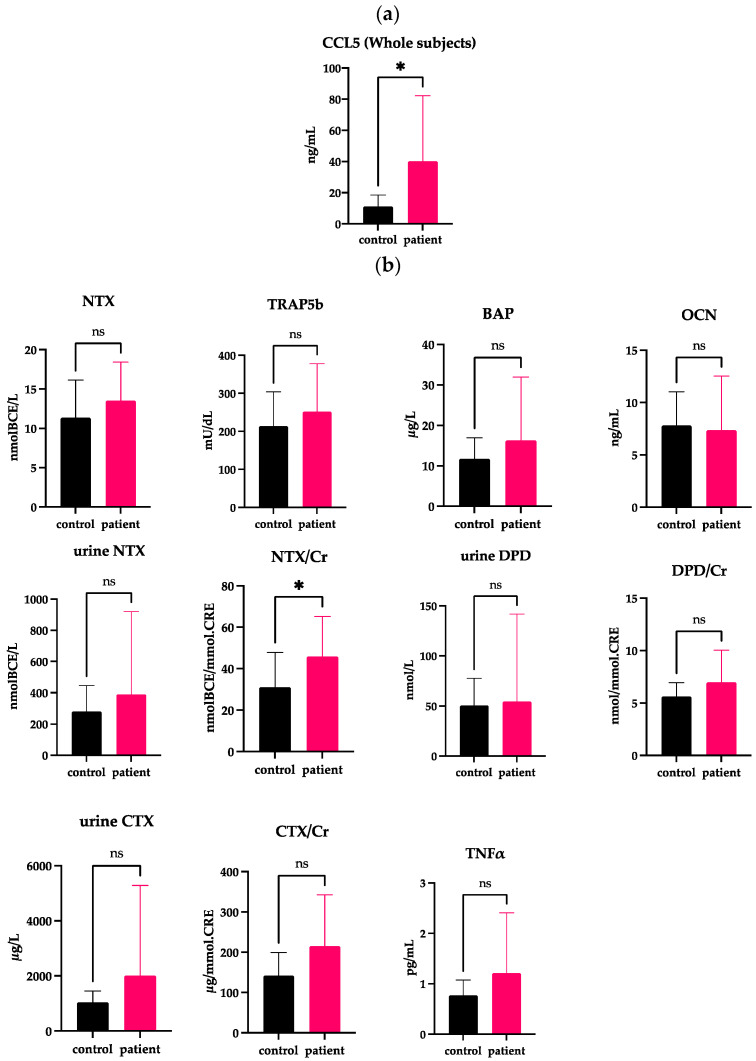
Measurement of CCL5 and other markers related to bone metabolism and inflammation in the serum or urine obtained from all subjects. (**a**) Serum CCL5 levels are significantly higher in the group of all DJD-TMJ subjects (n = 17) than in control groups (n = 17). (**b**) Serum or urine markers related to bone metabolism and inflammation were compared between the control group (n = 17) and the group of all DJD-TMJ subjects (n = 17). *t*-test, two-tailed. * Indicates a significant difference at *p* < 0.05.

**Figure 2 ijms-24-02775-f002:**
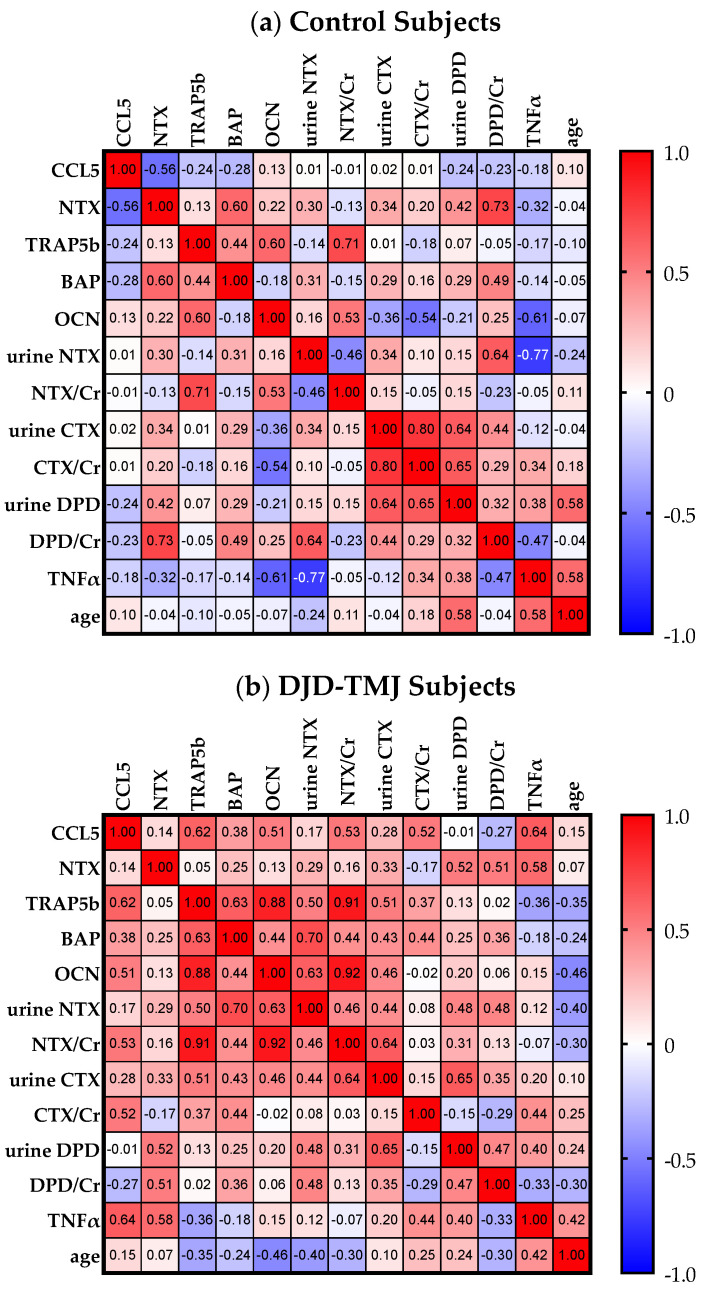
Multivariate analyses between molecules in (**a**) the control groups (n = 17) and (**b**) the group of all DJD-TMJ subjects (n = 17). The color depth of red and blue indicates the degree of positive and negative associations, respectively, between indicated molecules.

**Figure 3 ijms-24-02775-f003:**
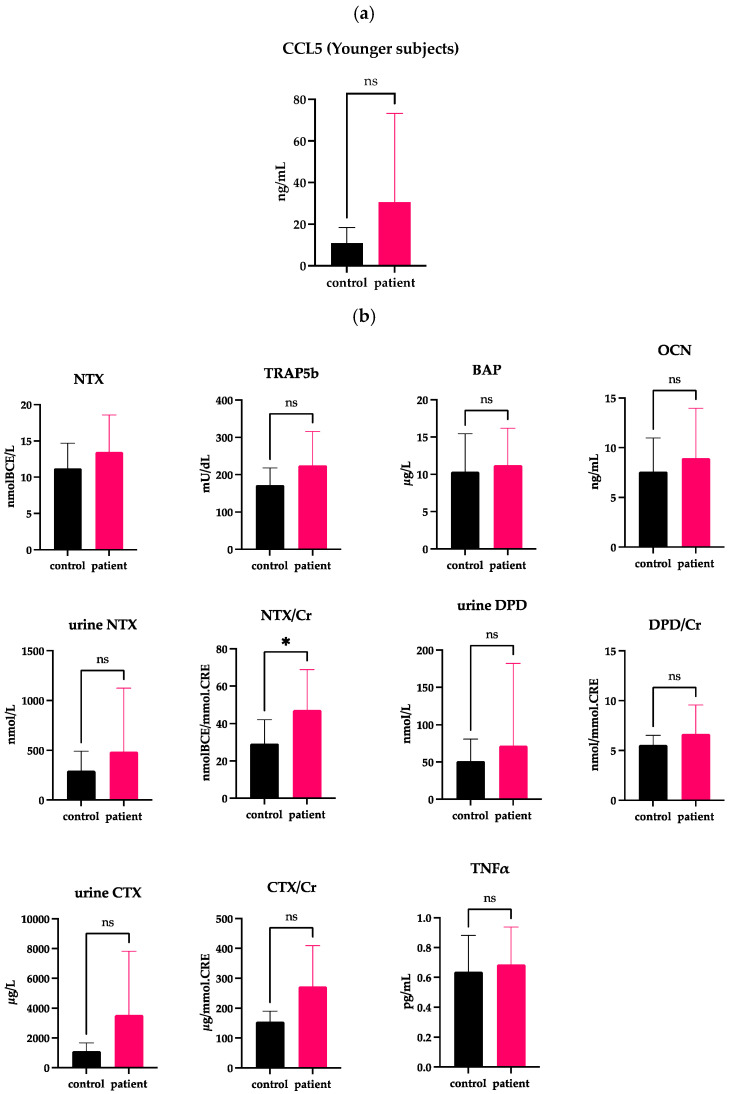
Measurement of CCL5 and other markers in the serum or urine obtained from subjects at ≤42 years old (Younger subjects). (**a**) Serum CCL5 levels are significantly higher in the group of DJD-TMJ subjects ≤42 years old (n = 10) than in the age-matched control groups (n = 11). (**b**) Serum or urine markers related to bone metabolism and inflammation were compared between the corresponding control groups (n = 11) and the group of DJD-TMJ subjects ≤ 42 years old (n = 10). *t*-test, two-tailed. * Indicates a significant difference at *p* < 0.05.

**Figure 4 ijms-24-02775-f004:**
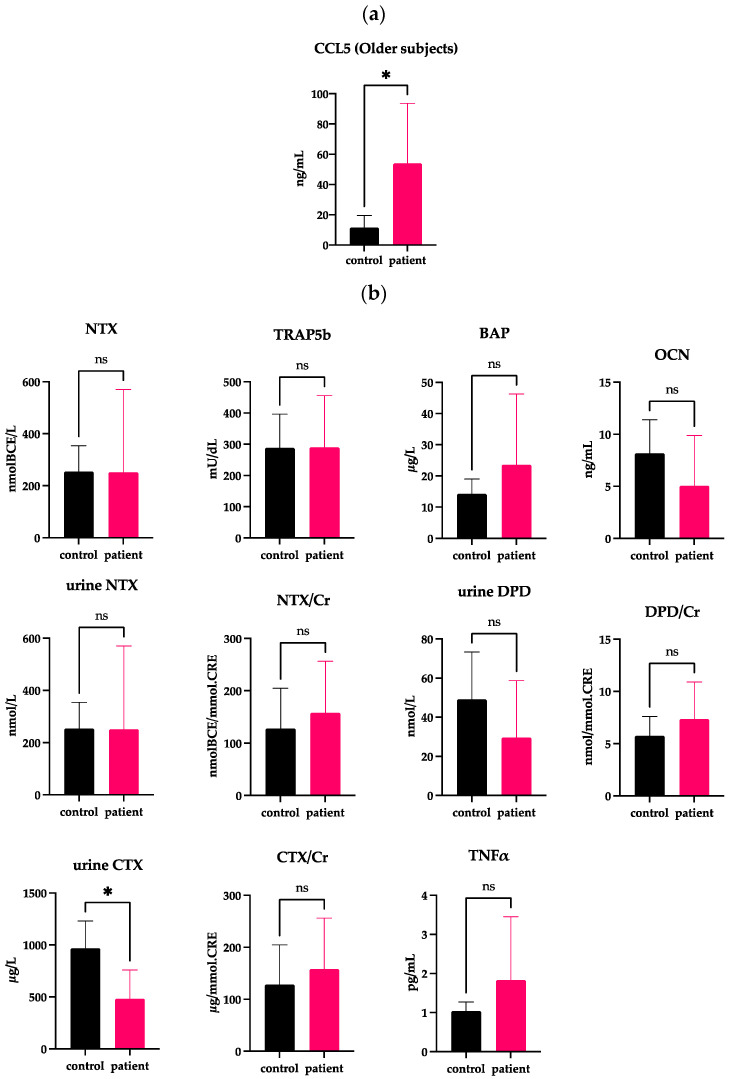
Measurement of CCL5 and other markers in the serum or urine obtained from subjects >42 years old (Older subjects). (**a**) Serum CCL5 levels are significantly higher in the group of DJD-TMJ subjects > 42 years old (n = 7) than in the age-matched control groups (n = 6). (**b**) Serum or urine markers related to bone metabolism and inflammation were compared between the corresponding control groups (n = 7) and the group of DJD-TMJ subjects >42 years old (n = 6). *t*-test, two-tailed. * indicates a significant difference at *p* < 0.05.

**Figure 5 ijms-24-02775-f005:**
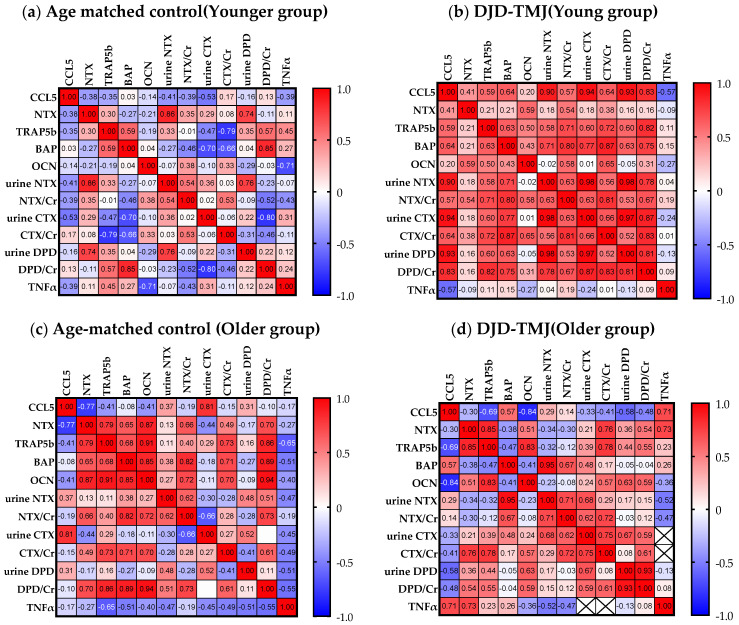
Multivariate analyses between molecules in (**a**) the control group at ≤42 years old (n = 11) and (**b**) the group of all DJD-TMJ subjects at ≤42 years old (n = 10), (**c**) the control group at >42 years old (n = 6) and (**d**) the group of all DJD-TMJ subjects at >42 years old (n = 7). The color depth of red and blue indicates the degree of positive and negative associations, respectively, between indicated molecules.

**Figure 6 ijms-24-02775-f006:**
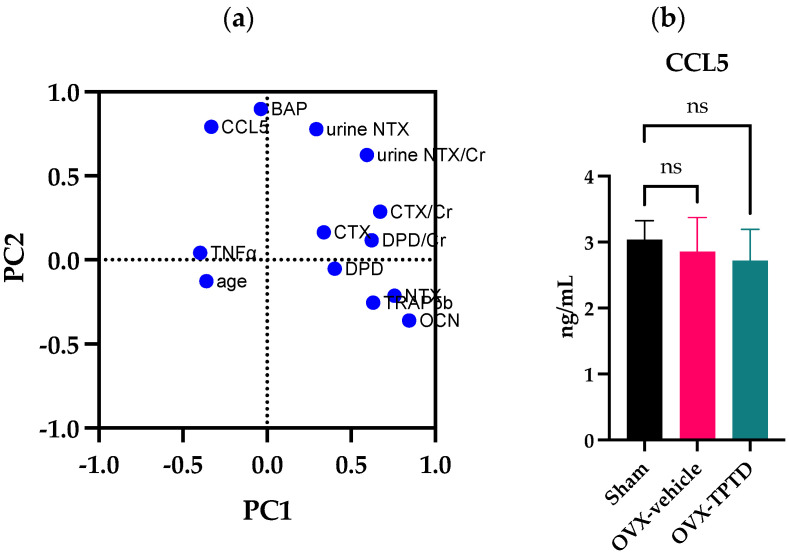
(**a**) A principal component analysis of all control and DJD-TMJ subjects and (**b**) the measurement of serum CCL5 levels in an ovariectomized (OVX) rat model. (**a**) The principal component analysis showed that CCL5, bone metabolism markers, and TNF-α were independent. (**b**) Serum CCL5 levels in sham-operated, OVX and OVX-TPTD-treated rats did not differ significantly. Dunnett’s test.

**Figure 7 ijms-24-02775-f007:**
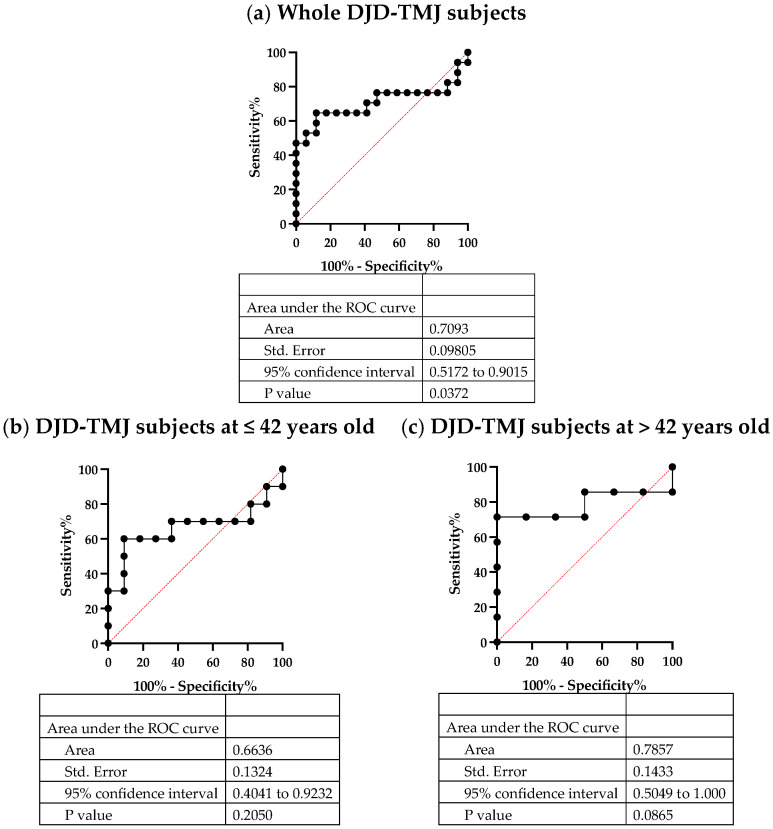
ROC curve analyses of CCL5. ROC curve analyses of CCL5 in (**a**) all DJD-TMJ patients (n = 17), (**b**) the group of all DJD-TMJ subjects ≤ 42 years old (n = 11), (**c**) the group of all DJD-TMJ subjects > 42 years old (n = 7) are shown.

**Figure 8 ijms-24-02775-f008:**
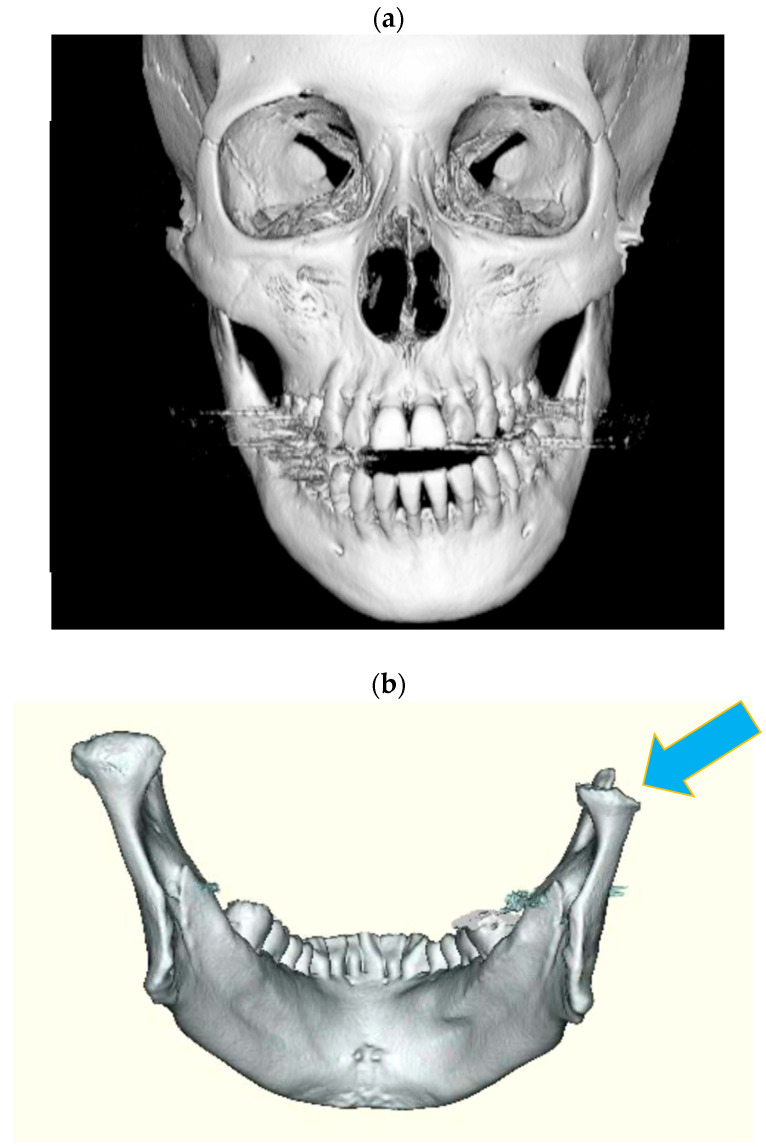
CT photographs of representative DJD-TMJ patients. (**a**) The anterior view of a 3D-reconstructed image of the CT showed the resorption of the left condyle and the mandibular deviation toward the left. (**b**) The posterior view of the 3D reconstructed mandible of another DJD-TMJ patient showed severe resorption in the right mandibular condyle (indicated by an arrow).

**Table 1 ijms-24-02775-t001:** List of laboratory results for all DJD-TMJ and control subjects. Highlighted in yellow shows patients with systemic diseases.

		Age	Sex	CCL5	NTX	Urine NTX	Urine NTX/Cr	CTX	CTX/Cr	DPD	DPD/Cr	BAP	TRAP5b	OCN	TNFα
**patient**	**1**	10’s	F	146.00	17.2	2155.0	74.9	10,691.0	373.0	377.0	13.1	18.7	375.0	8.3	
**patient**	**2**	10’s	F	27.90	9.9	957.0	71.1	4334.0	314.0	100.5	7.5	20.0	221.0	9.8	
**patient**	**3**	20’s	F	31.58	19.2	225.0	75.4	1215.0	414.0	26.6	8.9	15.5	381.0	20.6	0.82
**patient**	**4**	20’s	F	11.30	11.9	69.0	34.3			12.1	6.1	10.8	205.0	8.5	0.80
**patient**	**5**	20’s	F	2.76	10.9	256.0	29.9			34.1	4.0	7.3	147.0	4.9	1.0
**patient**	**6**	20’s	F	1.59	7.4	310.0	28.1			49.7	4.5	6.6	253.0	5.9	0.50
**patient**	**7**	20’s	F	3.40	10.2	336.0	51.0			49.1	7.5	7.9	233.0	4.3	
**patient**	**8**	30’s	F	16.50	9.2	37.0	12.7			16.3	5.6	8.7	136.0	4.6	0.70
**patient**	**9**	30’s	F	42.57	15.7	241.0	41.7	958.0	167.0	38.1	6.6	8.6	168.0	13.8	0.3
**patient**	**10**	30’s	F	21.58	23.2	282.0	52.3	498.0	93.0	16.3	3.0	8.3	133.0	8.9	
**patient**	**11**	50’s	F	96.40	9.5	957.0	71.1	785.0	206.0	28.1	7.3	74.3	139.0	1.0	
**patient**	**12**	50’s	F	36.40	8.6	183.0	38.8	492.0	103.0	30.3	6.4	11.6	150.0	1.2	1.0
**patient**	**13**	60’s	F	80.00	21.3	70.0	26.1			19.4	7.3	15.1	396.0	2.7	4.70
**patient**	**14**	60’s	F	100.00	11.1	51.0	48.8	100.0	97.0	6.1	5.8	21.1	140.0	1.3	
**patient**	**15**	60’s	F	10.20	15.9	133.0	58.3	699.0	309.0	15.1	6.6	10.5	506.0	10.0	1.0
**patient**	**16**	70’s	F	2.12	18.6	258.0	41.2			92.9	14.8	14.9	481.0	13.0	1.5
**patient**	**17**	70’s	F	52.30	10.2	110.0	24.7	341.0	75.0	15.2	3.4	17.6	216.0	6.2	1.00
**control**	**1**	10’s	F	12.70	10.0	91.0	18.8	502.0	107.0	38.4	7.9	22.5	243.0	8.1	
**control**	**2**	20’s	F	11.20	17.1	383.0	41.7	1346.0	149.0	44.7	4.9	11.0	174.0	11.0	0.4
**control**	**3**	20’s	F	9.80	16.6	396.0	28.7	1934.0	143.0	67.6	4.9	6.4	124.0	11.6	0.30
**control**	**4**	20’s	F	16.18	8.1	143.0	32.7	854.0	201.0	23.4	5.4	6.9	121.0	9.2	0.49
**control**	**5**	20’s	F	5.28	14.0	223.0	40.0	947.0	175.0	32.3	5.8	10.7	151.0	13.7	0.49
**control**	**6**	20’s	F	3.19	12.8	67.0	22.9			13.8	4.7	7.0	166.0	6.0	1.00
**control**	**7**	20’s	F	1.79	10.8	531.0	51.1			46.0	4.4	7.2	193.0	4.5	0.80
**control**	**8**	30’s	F	9.65	7.4	289.0	6.4			77.3	6.4	17.3	203.0	5.3	1.00
**control**	**9**	30’s	F	14.50	8.5	341.0	22.1			84.6	5.5	8.7	128.0	2.9	0.70
**control**	**10**	40’s	F	28.80	8.0	109.0	21.1			26.8	5.2	8.0	142.0	5.2	0.50
**control**	**11**	40’s	F	6.65	10.2	679.0	37.8			109.0	6.1	8.3	250.0	6.2	0.70
**control**	**12**	50’s	F	3.83	22.3	357.0	77.9	577.0	128.0	35.3	7.7	20.0	352.0	12.0	1.00
**control**	**13**	50’s	F	26.10	2.4	343.0	38.7			50.5	5.7	14.8	191.0	6.4	1.1
**control**	**14**	70’s	F	5.96	8.1	155.0	12.0	850.0	65.0	40.7	3.2	6.8	182.0	5.0	1.4
**control**	**15**	70’s	F	12.20	8.1	177.0	19.2	1015.0	109.0	37.1	4.0	12.8	210.0	5.2	0.90
**control**	**16**	70’s	F	13.30	12.0	335.0	22.4	1222.0	82.0	97.0	6.5	12.4	361.0	8.5	0.80
**control**	**17**	70’s	F	8.08	16.6	157.0	34.4	1175.0	258.0	34.0	7.5	18.7	436.0	12.0	

**Table 2 ijms-24-02775-t002:** List of laboratory results for DJD-TMJ and control subjects at ≤42 years old (Younger subjects). Highlighted in yellow shows patients with systemic diseases.

		Age	Sex	CCL5	NTX	Urine NTX	Urine NTX/Cr	CTX	CTX/Cr	DPD	DPD/Cr	BAP	TRAP5b	OCN	TNF-α
**patient**	**1**	10’s	F	146.00	17.2	2155.0	74.9	10,691.0	373.0	377.0	13.1	18.7	375.0	8.3	
**patient**	**2**	10’s	F	27.90	9.9	957.0	71.1	4334.0	314.0	100.5	7.5	20.0	221.0	9.8	
**patient**	**3**	20’s	F	31.58	19.2	225.0	75.4	1215.0	414.0	26.6	8.9	15.5	381.0	20.6	0.82
**patient**	**4**	20’s	F	11.30	11.9	69.0	34.3			12.1	6.1	10.8	205.0	8.5	0.80
**patient**	**5**	20’s	F	2.76	10.9	256.0	29.9			34.1	4.0	7.3	147.0	4.9	1.0
**patient**	**6**	20’s	F	1.59	7.4	310.0	28.1			49.7	4.5	6.6	253.0	5.9	0.50
**patient**	**7**	20’s	F	3.40	10.2	336.0	51.0			49.1	7.5	7.9	233.0	4.3	
**patient**	**8**	30’s	F	16.50	9.2	37.0	12.7			16.3	5.6	8.7	136.0	4.6	0.70
**patient**	**9**	30’s	F	42.57	15.7	241.0	41.7	958.0	167.0	38.1	6.6	8.6	168.0	13.8	0.3
**patient**	**10**	30’s	F	21.58	23.2	282.0	52.3	498.0	93.0	16.3	3.0	8.3	133.0	8.9	
**control**	**1**	10’s	F	12.70	10.0	91.0	18.8	502.0	107.0	38.4	7.9	22.5	243.0	8.1	
**control**	**2**	20’s	F	11.20	17.1	383.0	41.7	1346.0	149.0	44.7	4.9	11.0	174.0	11.0	0.4
**control**	**3**	20’s	F	9.80	16.6	396.0	28.7	1934.0	143.0	67.6	4.9	6.4	124.0	11.6	0.30
**control**	**4**	20’s	F	16.18	8.1	143.0	32.7	854.0	201.0	23.4	5.4	6.9	121.0	9.2	0.49
**control**	**5**	20’s	F	5.28	14.0	223.0	40.0	947.0	175.0	32.3	5.8	10.7	151.0	13.7	0.49
**control**	**6**	20’s	F	3.19	12.8	67.0	22.9			13.8	4.7	7.0	166.0	6.0	1.00
**control**	**7**	20’s	F	1.79	10.8	531.0	51.1			46.0	4.4	7.2	193.0	4.5	0.80
**control**	**8**	30’s	F	14.50	8.5	341.0	22.1			84.6	5.5	8.7	128.0	2.9	0.70
**control**	**9**	30’s	F	9.65	7.4	289.0	6.4			77.3	6.4	17.3	203.0	5.3	1.00
**control**	**10**	40’s	F	28.80	8.0	109.0	21.1			26.8	5.2	8.0	142.0	5.2	0.50
**control**	**11**	40’s	F	6.65	10.2	679.0	37.8			109.0	6.1	8.3	250.0	6.2	0.70

**Table 3 ijms-24-02775-t003:** The list of laboratory results for DJD-TMJ and control subjects at >42 years old (Older subjects). Highlighted in yellow shows patients with systemic diseases.

		Age	Sex	CCL5	NTX	Urine NTX	Urine NTX/Cr	CTX	CTX/Cr	DPD	DPD/Cr	BAP	TRAP5b	OCN	TNF-α
**patient**	**1**	50’s	F	96.40	9.5	957.0	71.1	785.0	206.0	28.1	7.3	74.3	139.0	1.0	
**patient**	**2**	50’s	F	36.40	8.6	183.0	38.8	492.0	103.0	30.3	6.4	11.6	150.0	1.2	1.0
**patient**	**3**	60’s	F	80.00	21.3	70.0	26.1			19.4	7.3	15.1	396.0	2.7	4.7
**patient**	**4**	60’s	F	100.00	11.1	51.0	48.8	100.0	97.0	6.1	5.8	21.1	140.0	1.3	
**patient**	**5**	60’s	F	10.20	15.9	133.0	58.3	699.0	309.0	15.1	6.6	10.5	506.0	10.0	1.0
**patient**	**6**	70’s	F	2.12	18.6	258.0	41.2			92.9	14.8	14.9	481.0	13.0	1.5
**patient**	**7**	70’s	F	52.30	10.2	110.0	24.7	341.0	75.0	15.2	3.4	17.6	216.0	6.2	1.00
**control**	**1**	50’s	F	3.83	22.3	357.0	77.9	577.0	128.0	35.3	7.7	20.0	352.0	12.0	1.00
**control**	**2**	50’s	F	26.10	2.4	343.0	38.7			50.5	5.7	14.8	191.0	6.4	1.1
**control**	**3**	70’s	F	5.96	8.1	155.0	12.0	850.0	65.0	40.7	3.2	6.8	182.0	5.0	1.4
**control**	**4**	70’s	F	12.20	8.1	177.0	19.2	1015.0	109.0	37.1	4.0	12.8	210.0	5.2	0.90
**control**	**5**	70’s	F	13.30	12.0	335.0	22.4	1222.0	82.0	97.0	6.5	12.4	361.0	8.5	0.80
**control**	**6**	70’s	F	8.08	16.6	157.0	34.4	1175.0	258.0	34.0	7.5	18.7	436.0	12.0	

## Data Availability

Supplementary data are available at https://www.mdpi.com/article/10.3390/ijms24032775/s1.

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
