# Peer review of "Association between an Increased Serum CCL5 Level and Pathophysiology of Degenerative Joint Disease in the Temporomandibular Joint in Females"

_ijms, 2023, doi:10.3390/ijms24032775_

Round 1
Reviewer 1 Report
Thank you for the opportunity to read the manuscript "Association between an increased serum CCL5 level and pathophysiology of condylar resorption in temporomandibular joint in female" before publication. Below are my comments, which I hope will prove helpful to the authors:
Abstract:
- CCL5 abbreviation should be explained on first use.
Introduction:
- The division of DJD-TMJ into subtypes is a delicate matter, different centers adopt different classifications. Give references to the presented classification or write clearly that you propose such a division for the purposes of this study.
Materials and methods:
- I have no comments in this section.
Results:
- I have no comments in this section.
Discussion:
- I have no comments in this section.
Back Matter:
- I have no comments in this section.
Reference:
- Delete items older than 10 years unless they are necessary.
Overall, I consider this manuscript to be a well-written report of a reasonably designed study with clearly stated limitations and conclusions.
Reviewer 2 Report
Thank you for your submission the International Journal of Molecular Sciences
The 3rd paragraph of the introduction is slightly confusing as it jumps from subtypes of DJD (2 of which are condylar resorption related and 1 completely different = OA) to the etiology of TMJ-DJD and then back to the prevalence of ICR/PCR. This follows through later where you talk about the pathophysiology of DJD (L81) and then go back to IDR/PCR (L98). ICR/PCR is considered by most to be a different disease to advanced DJD seen with OA and other inflammatory joint diseases, and therefore as your study seems to include patients of all ages and medical conditions, then you should probably omit most reference to ICR/PCR and OA and characterise your study as assessing patients with advanced degenerative changes of the TMJ, with multiple different etiologies (and all of these conditions should be in the discussion about possible etiologies, but not part of the introduction).
The point that the serum CCL5 was not as raised in younger patients is worth discussing further, as these are the patients most likely to have classic ICR/PCR. This also impacts on your conclusions, as based on your statement that the serum CCL5 does not in fact reach significantly higher levels in patients than it does in controls, you can't really say this is an effective biomarker of ICR/POCR in this age group.
Some of the terminology is also not really correct - eg incident rate (do you mean incidence cases / per population / per time frame, or prevalence ( cases per population either at defined time or age etc)?
You can't really say the " Conclusively" (L 360) CCL5 "might" be an effective biomarker - if it's conclusive it either is effective or it is not effective.
I think the experimental model itself is reasonable and seems to cover the options that need addressing to draw any conclusions, it's just a pity the the clinical groups are not properly separated and clarified
Round 2
Reviewer 2 Report
Thank you for this revised manuscript, which is much improved